# Are we walking the talk of participatory Indigenous health research? A scoping review of the literature in Atlantic Canada

**Kathleen Murphy**[1], **Karina Branje**[1], **Tara White**[1], **Ashlee Cunsolo**[2], **Margot Latimer**[1],
**Jane McMillan**[3], **John R. Sylliboy**[4], **Shelley McKibbon**[5], **Debbie Martin**[1] *

**1** Faculty of Health, Dalhousie University, Halifax, Nova Scotia, Canada, **2** School of Arctic and Sub-Arctic Studies, Labrador Institute of Memorial University, Memorial University, Happy Valley-Goose Bay, Newfoundland & Labrador, Canada, **3** Department of Anthropology, St. Francis Xavier University, Antigonish, Nova Scotia, Canada, **4** Department of Integrated Studies in Education (DISE), McGill University, Montreal, Quebec, Canada, **5** Kellogg Health Sciences Library, Dalhousie University, Halifax, Nova Scotia, Canada

* debbie.martin@dal.ca

## Abstract

### Introduction

Participatory research involving community engagement is considered the gold standard in Indigenous health research. However, it is sometimes unclear whether and how Indigenous communities are engaged in research that impacts them, and whether and how engagement is reported. Indigenous health research varies in its degree of community engagement from minimal involvement to being community-directed and led. Research led and directed by Indigenous communities can support reconciliation and reclamation in Canada and globally, however clearer reporting and understandings of community-led research is needed. This scoping review assesses (a) *how* and *to what extent* researchers are reporting community engagement in Indigenous health research in Atlantic Canada, and (b) what recommendations exist in the literature regarding participatory and community-led research.

### Methods

Eleven databases were searched using keywords for Indigeneity, geographic regions, health, and Indigenous communities in Atlantic Canada between 2001-June 2020. Records were independently screened by two reviewers and were included if they were: peer-reviewed; written in English; health-related; and focused on Atlantic Canada. Data were extracted using a piloted data charting form, and a descriptive and thematic analysis was performed. 211 articles were retained for inclusion.

### Results

Few empirical articles reported community engagement in all aspects of the research process. Most described incorporating community engagement at the project's onset and/or during data collection; only a few articles explicitly identified as entirely community-directed or led. Results revealed a gap in reported capacity-building for both Indigenous communities

**Data Availability Statement:** All relevant data are within the paper and its Supporting Information files.

**Funding:** Author Initials: AC, LJM, ML, DM; Grant #: NEH-160643; Program: CIHR Operating Grant: Network Environments for Indigenous Health Research - Development Grants; Name of Funder: Canadian Institutes of Health Research (CIHR); URL of CIHR: https://cihr-irsc.gc.ca/e/193.html The funders did not play a role in study design, data collection, and analysis, the decision to publish, or the preparation of the manuscript. Author initials: DM; Name of Funder: Government of Canada, Canada Research Chairs; Program Title: Canada Research Chair, Tier II Canada Research Chair in Indigenous Peoples' Health and Well-Being; URL: https://www.chairs-chaires.gc.ca/home-accueil-eng.aspx The funders did not play a role in study design, data collection, and analysis, the decision to publish, or the preparation of the manuscript.

**Competing interests:** The authors have declared that no competing interests exist.

and researchers, necessary for holistic community engagement. Also revealed was the need for funding bodies, ethics boards, and peer review processes to better facilitate participatory and community-led Indigenous health research.

## Conclusion

As Indigenous communities continue reclaiming sovereignty over identities and territories, participatory research must involve substantive, agreed-upon involvement of Indigenous communities, with community-directed and led research as the ultimate goal.

## Introduction

Indigenous health research conducted "in a good way" has the potential to support Indigenous reconciliation and reclamation in Canada [1, 2 (p. 33)]. Foundational to achieving substantive reconciliation is supporting the self-determination of Indigenous Peoples [3] and honouring distinct Indigenous knowledges and ways of knowing [4]. Indeed, First Nations, Inuit, and Métis in Canada have long been utilizing their traditional and cultural knowledge to meet their needs and priorities through complex methods of inquiry [5, 6]; yet Western academic research systems and approaches have historically excluded and/or appropriated Indigenous voices, with research largely being conducted *on* Indigenous communities rather than with, for, or by them [7]. Essentially, a great deal of academic research has furthered the colonial agenda rather than questioned or denounced it. However, Indigenous communities throughout Canada and around the world are transforming health research in ways that respond to community priorities, contribute to Indigenous self-determination, and insist on research that is led and governed by Indigenous Peoples [8–12]. This is the essence of community-driven Indigenous health research [13].

With a growing number of Indigenous researchers entering the academy [14], academic research is increasingly recognizing the complexities resulting from colonization, assimilation, racism, and oppression, and the need to support Indigenous Peoples' involvement in research to navigate these complexities in a respectful and culturally sensitive way [15, 16]. Research that is participatory is often cited as a means to navigate these complexities, and is a way of undertaking research that encompasses a number of approaches, including community-based participatory research (CBPR), participatory action research (PAR), decolonizing methodologies, and others [17]. For our purposes, we are broadly defining participatory research as that which engages communities throughout the research process, as we recognize that not all studies formally define their approach. It is premised on characteristics that involve shifting the balance of power from researchers to the community [18–20], co-learning and capacity building for all partners involved [21], and repositioning "scholars as participants of a process in which they listen, learn, and offer service" [13 (p. 4), 19, 21–23]. Key to participatory research is community engagement, or the inclusive participation of community members throughout the research process, ranging from relationship building before the research takes place and the identification of a research question, through to data collection, analysis, and the sharing and uptake of results [24, 25]. Rather than a linear researcher-participant relationship, participatory and community-driven research encourages scholars and community members to enter into an authentic partnership to achieve shared goals [21, 26]. It requires flexibility, enabling research teams to pivot based on emerging needs and concerns of community partner(s) [27, 28], and thus the unique context in which the research takes place is of critical

importance [13]. Importantly, there is no 'one size fits all' approach [29]. Although the movement towards participatory research is arguably a far cry from the damaging effects of past research that has had no Indigenous involvement, it has nonetheless been recognized as a promising approach to achieving Indigenous self-determination through research [18, 30]. However, implementing participatory principles is not always straight-forward in practice [13, 30–32], and is not always evident in published research.

There is a spectrum of community engagement in Indigenous health research [21, 25, 33, 34]. At one end, a growing body of research is community-initiated and undertaken, however at the other end, there remains a breadth of largely academic-driven research with minimal Indigenous community involvement [13]. While it is important that participatory research be context-specific and reflect the unique needs and preferences of the communities involved [20, 33], this spectrum of community engagement raises questions regarding how to define and accurately measure the extent to which a research project is, in fact, participatory, and how to measure its degree of community engagement [35]. If research fails to appropriately engage Indigenous communities, it may not only compromise the potential benefits of the research for the Indigenous Peoples, communities, governments, and organizations involved, but it may also risk perpetuating the colonial narratives that this type of research is striving to contradict [13].

In Canada, we have witnessed a cascade of initiatives aiming to build capacity for Indigenous health research. The most recent example is an investment by the Canadian Institutes for Health Research to launch nine Network Environments for Indigenous Health Research (NEIHR) across Canada. These NEIHRs are intended to facilitate research that is "driven by, and grounded in Indigenous communities in Canada" [36 (para.2)]. In Atlantic Canada, the goal of the Wabanaki-Labrador NEIHR (WLN) is to support community-driven research and to shift research directly into the hands of Indigenous communities, and to build capacity to enable this transformation to happen. To inform the development and implementation of the WLN, we undertook this scoping review to understand the range, nature, and characteristics of health research conducted in our region over the last two decades (see White et al. [34] for a descriptive summary of our results). A key finding from our previous analysis was that, despite the recognized importance of participatory research that shifts power and control to Indigenous communities and organizations [37], the reported levels of community engagement over the twenty-year period remained virtually unchanged. Thus, the present research is interested in seeing *whether*, *how*, and *to what extent* research teams are reporting participatory and/or Indigenous-led research, by analyzing the reported methods of community engagement and how they align with recommendations in academic, peer-reviewed literature.

Atlantic Canada is a unique region for more deeply understanding the nature and extent of community-engaged Indigenous health research, as the territory is home to diverse groups of Indigenous Peoples including Inuit, Innu, Mi'kmaq, Wolastiqiyik, and Passamaquoddy, all of whom have distinct cultures, histories and languages. Within this diverse region, there is a growing capacity to assess, review and engage with research, reflected in the number of regional and local research review processes and robust guidelines for community-directed and participatory research with Indigenous Peoples (e.g. Mi'kmaw Ethics Watch, the Nunatsiavut Government, the NunatuKavut Community Council, the Mi'kmaq Confederacy of PEI, and the Native Council of PEI, with others currently in development). We also acknowledge the tremendous value in undertaking place-based research for Indigenous communities, evidenced by Indigenous health research projects grounded in context, place, and on-the-land engagement [38–40], and the difficulty of approaching Indigenous health using a pan-Indigenous lens. Thus, we assert that a geographically-bounded scoping review offers a unique way to provide insight into Indigenous health research engagement within the specific region of

Atlantic Canada. This process of establishing foundations for placed-based, Indigenous health research mirrors the processes of self-determination, decolonization, and self-governance regionally, nationally, and internationally by Indigenous Peoples.

To better understand whether researchers are 'walking the talk' of participatory Indigenous health research in Atlantic Canada, we sought to uncover: (a) how researchers are reporting community engagement in Indigenous health research, and (b) what recommendations exist in the literature with regard to participatory and community-led research.

## Methods

We utilized the scoping review methods proposed by Arksey and O'Malley [41] and advanced by Levac, Colguhoun and O'Brien [42]. We established and collaborated with an Indigenous Advisory Committee, with whom we met with twice at the outset of this review to conceptualize and develop the search strategy, and to ensure that the research questions and search strategies aligned with the knowledge and goals of end-users. The Indigenous Advisory Committee (IAC) was composed of Indigenous individuals from organizations and governing bodies that represent the diversity of Indigenous communities throughout the Atlantic region. We also held a Co-Learning Health Research Summit in June 2019, where we presented the scoping review process to Indigenous and allied representatives of community-based organizations, governments, and academic institutions, at which time we received suggestions for how to strengthen the review.

### Search strategy

We worked with subject experts, an experienced librarian, and the Indigenous Advisory Committee, to create a list of specific keyword terms for the people and places/communities that are the focus of this review. These specific keywords fell under three broad concepts of interest: (1) "Indigenous Peoples in Atlantic Canada", (2) "geographic regions and Indigenous communities in Atlantic Canada", and for databases not related to health, controlled vocabulary terms for the concept of (3) "health research" was included (see S1 Table). Because there was no single subject heading for the concept of "health research" and multiple potentially relevant headings for different types of research, all available research headings were used. The "health research" concept was therefore represented by a search for either the health-related keywords or a research-related subject heading. Finally, given their publishing record in academic journals, Native Friendship Centres were also included in the search strategy, as a keyword under the broad concept of "geographic regions and Indigenous communities in Atlantic Canada". To inform the search terms used, we consulted the University of Alberta Libraries search term filters for Indigenous Peoples [43], the Mi'kmaw Resource Centre at the Unama'ki College, Cape Breton University [44], and relied on knowledge from research team members who have long-standing relationships with a number of Indigenous communities in the region. These terms were used to search each database. In addition, the controlled vocabulary of each database was searched to find subject terms equivalent to the people/community keywords.

### Information sources

Nine databases were searched for published literature, including Academic Search Premier, Bibliography of Native North Americans, CAB Abstracts, CINAHL, Embase, PsycINFO, Public Affairs Index, PubMed, and Web of Science. Scopus and ProQuest Dissertations & Theses Global were also searched for theses and dissertations. The search strategy was peer-reviewed by health science librarians before the searches were conducted. Database searches were

completed by May 2020. No date or language limits were applied. References were then imported into EndNote for removal of duplicates, and finally into Covidence for screening.

## Selection of sources of evidence and eligibility

The titles and abstracts of all articles were independently screened by two members of our research team based on whether the studies: (1) focused on an Indigenous community/population; (2) took place, or drew conclusions specific to Indigenous Peoples in Atlantic Canada; (3) were health research (inclusive of the upstream social determinants of health); (4) were published between January 2001 –June 2020; and (5) were written in the English language (Table 1). The exclusion criteria included: books, book chapters, and grey literature. During both the title/abstract and full-text screening phases, the research team met at the beginning, mid-point, and final stages of reviewing to discuss challenges and questions encountered, and to ensure consistency in screening. If there was a discrepancy among reviewers as to whether the full text ought to be reviewed, a third reviewer assessed the article for inclusion.

## Data charting process and data items

A data charting form was collaboratively developed by all members of our research team (S2 Table). Two researchers piloted the form by independently extracting 10 randomized articles, and then reviewing their extraction with the full research team to (1) ensure that researchers were consistent in how they charted the data; and (2) modify the data charting form where necessary. The following data were extracted: publication date; context of research (Indigenous nation/population, target communities, geography, purpose); health topics covered; research type; methodology, ontology and epistemology used; institutional and Indigenous ethics approval; whether a community researcher was listed as a co-author; level of community engagement; and recommendations/implications of the research.

## Synthesis of results

Data analysis involved a descriptive numerical summary of the quantitative data, as well as a deductive and inductive thematic content analysis of the qualitative data to identify themes of analytic interest [46]. The quantitative analysis included empirical articles only and is reported elsewhere [34], while the qualitative analysis included both empirical and non-empirical articles. Initial codes were developed inductively by the research team within each category of

**Table 1. Summary of inclusion criteria used to identify Indigenous health research in Atlantic Canada between 2001-June 2020 (adapted from Jones et al., 2018) [45].**

| INCLUSION CRITERIA | | LEVEL 1 SCREENING: Titles and Abstracts | LEVEL 2 SCREENING: Full Text |
|---|---|---|---|
| Population Criteria | Included an Indigenous population that is identifiable at an individual level or by geographic proxy | X | X |
| Geographic Criteria | Explicitly examined populations, communities, or regions in Atlantic Canada | X | X |
| | National and/or international studies if conclusions are drawn specific to the Atlantic region (i.e. substantial component of research focuses on Atlantic region) | | X |
| Study Design Criteria | Be primary research, reviews, or other scholarly articles (e.g. commentaries, perspectives, and opinions) | | X |
| Human Health Criteria | Be focused on a health-related topic or issue | X | X |
| Language and Date Criteria | Be published between January 2001 to June 2020 | X | X |
| | An English version of the title/ and or abstract was available | X | X |

extraction and refined upon consensus throughout the iterative and organic coding process. Once data were coded, it was collated and organised into themes. Themes were then reviewed against the data to discern their appropriateness and validity, and refined further where necessary [46].

## Ethical considerations

Ethics approval was not required for this review; however, to ensure our research was respectful, relevant, reciprocal, and responsible [47], we sought feedback from our IAC as well as other Indigenous rightsholders at various points throughout the scoping review process. This collaborative approach was important for privileging Indigenous knowledges and ways of knowing which are grounded in 'place'. The origin stories of Indigenous Peoples and all of the subsequent knowledge that grounds the array of Indigenous cultures is rooted in locality, for example "Mi'kmaq is so intimately tied to a collective identity, to one's relationship with the self, others, and nature. . ." [48 (p. 16–17)]. Thus, we acknowledge the incredible diversity of Indigenous populations within Atlantic Canada (i.e., Inuit, Innu, Mi'kmaq, Wolastoqiyik, Passamaquoddy), and by engaging with community representatives at the outset of this review, we hope that the results are as useful as possible to all Indigenous groups in the region, in addition to elsewhere in Canada and globally.

## Results

### Selection of sources of evidence

Based on inclusion criteria (Table 1) 9056 articles were identified in the search (Fig 1), of which 211 (S1 File) were included in the qualitative analysis (empirical and non-empirical) and 185 articles (empirical) were included in the quantitative analysis. Three additional articles were identified through the screening process; two that arose as chapters in the same dissertation, and one that was identified in a reference list. For a descriptive overview of all studies reviewed, see White et al [34].

### Synthesis of results

**How was Indigenous health research conducted in Atlantic Canada?.** *Quantitative overview of community engagement.* The level of reported community engagement by year and

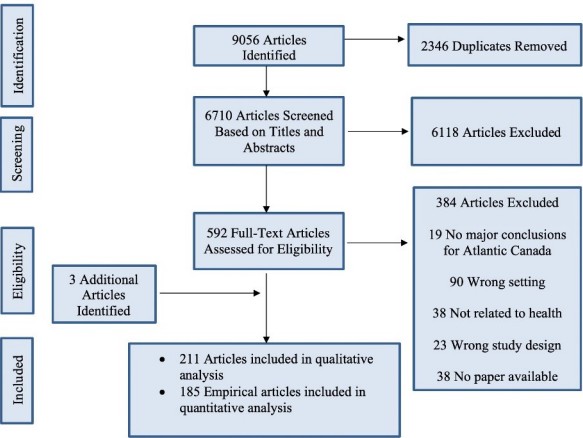

**Fig 1. Preferred reporting items for systematic review and meta-analysis (PRISMA) flow diagram** [49].

study design was explored for each article. Although several studies note the *importance* of engaging communities in all aspects of the research process, the majority of studies (n = 92; 49.7%) only reported engagement with communities in some aspects (i.e. development of research question, data collection, analysis, development of recommendations, and/ or dissemination), followed by studies not reporting community engagement at all (n = 46; 24.9%). Several studies (n = 37; 20.0%) also reported engagement with communities in one aspect of the research, and the fewest number of articles (n = 10; 5.4%) reported engagement throughout the entire research process (Table 2). Fig 2 demonstrates the percentage of studies that reported engagement with communities throughout key stages of the research process; including the development of the research question, data collection, data analysis, recommendations, and dissemination. Notably, when studies did report engagement with communities, the majority did so in the data collection phase (n = 95; 51.1%), with engagement during the development of the research question (n = 59; 31.7%) and data analysis (n = 52; 30.0%) following. Only 4 studies (1.9%) reported engagement with the community in the development of recommendations.

Our analyses also revealed that although the number of Indigenous health research articles published has increased from 2001–2020, the percentage of articles reporting Indigenous engagement in all or multiple aspects of the research process has remained consistent. While twenty-eight articles reported using participatory research, only 5 (17.9%) of those studies reported engaging the community in all aspects of the research process, and 20 (71.4%) articles reported engaging the community in two or more aspects of the research process. Of the 19 articles that reported using an Indigenous methodology, epistemology, or ontology to guide their research, 17 (89.5%) also reported engaging with the community at multiple or all stages of the research process.

*Qualitative overview of community engagement. Relationship building.* Researchers who described building relationships with communities did so in several ways. Some research was grounded in existing community relationships or longstanding partnerships, built over a number of years [50–57], and others were built on new relationships facilitated through community visits prior to initiating the research [58–60]. Relationship building was reported both as a formalized process, whereby a community oversight body (e.g., a community advisory committee, steering committee, working group) was created to direct the research [55, 61–80], and also through informal moments, such as researchers participating in community events and celebrations, including Powwows, community feasts, sea ice trips, strawberry socials, etc. [58, 67, 81–84]. One research team also hosted a homework club for children through which they were able to build trust and rapport with community members [67]. Many of these

**Table 2. Empirical articles that report engagement with community in various aspects of the research process.**

| | Total (n) | All aspects (n) | All aspects (%) | Two or more aspects (n) | Two or more aspects (%) | One aspect (n) | One aspect (%) | None (n) | None (%) |
|---|---|---|---|---|---|---|---|---|---|
| Total | 185 | 10 | 5.4% | 92 | 49.7% | 37 | 20.0% | 46 | 24.9% |
| 2001–2005 | 22 | 0 | 0.0% | 12 | 54.6% | 4 | 18.2% | 6 | 27.3% |
| 2006–2010 | 40 | 3 | 7.5% | 20 | 50.0% | 5 | 12.5% | 12 | 30.0% |
| 2011–2015 | 64 | 4 | 6.3% | 31 | 48.4% | 18 | 28.1% | 11 | 17.2% |
| 2016–2020 | 59 | 3 | 5.1% | 29 | 49.2% | 10 | 17.0% | 17 | 28.8% |
| CBPR | 28 | 5 | 17.9% | 20 | 71.4% | 2 | 7.1% | 1 | 3.6% |
| Indigenous methodology/ epistemology/ ontology | 19 | 3 | 15.9% | 14 | 73.7% | 0 | 0.0% | 2 | 10.5% |

Note. Aspects include development of research question, data collection, analysis, development of recommendations, and/or dissemination

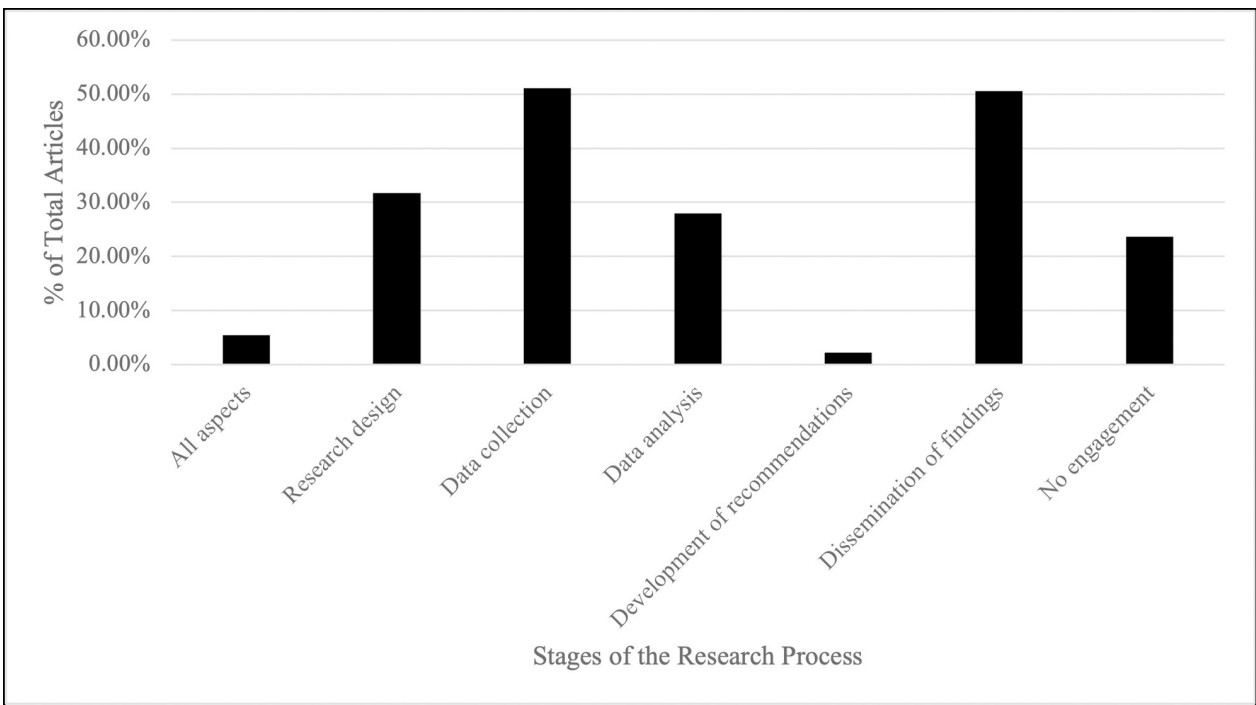

**Fig 2. Proportion of articles that reported engaging with the community at each aspect of the research process.**

relationship-building activities were reportedly undertaken in recognition of the importance of trust between researchers and communities [52, 57, 67, 85, 86], for instance as stated by Jardine and Furgal, "considerable time was invested in building relationships between the researchers and team members in the community to develop trust through a shared understanding of intentions, motivations, and interests" [57 (p. 122)]. One study, however, cautioned that while relationship building is critical for establishing trust and accountability, communicating the scope of this process in an academic paper "does not necessarily measure accountability or reciprocity on the part of the researcher, or indicate success from a community's perspective" [52 (p. 464)].

*Determination over the research process.* Indigenous or community advisory committees were often reported to offer guidance and input throughout the research process to ensure its relevance to communities, and in some cases studies were directly borne out of community requests for particular research projects [73, 74, 87–92]. Others reported being influenced by community-identified priorities [56, 58–60, 68, 85, 93–96]. Very few studies described that they were led entirely by the community; however, there were instances where community members were identified as co-decision makers and equal partners throughout the entirety of the project [67, 91, 97–99], at times being recognized as co-principal investigators and co-investigators in line with the tenets of participatory research [91, 98]. Yet, with the breadth of articles that made mention of their use of the principles of Ownership, Control, Access and Possession (OCAP™) in undertaking ethical Indigenous health research [51, 55, 80, 86, 90–92, 98–107], only one study explicitly gave recognition to communities as 'owners' of the research enterprise [67]. One study noted that the community and researchers wrote the grant proposal collaboratively. Similarly, only one study reported transferring all project finances directly to the community government: "due to an innovative funding model which saw all finances for this project transferred directly to the Rigolet Inuit Community Government, all research

decisions were co-decided between the community government and the research team" [97 (p. 17)]. In another instance, the community government was awarded the funding, and they convened a team of Indigenous and non-Indigenous researchers [108].

*Community engagement in specific aspects of the research process.* Reports of community engagement varied across studies. Of those studies that reported engaging communities during the design of the research process, they did so to ensure that project goals were in the interest of their partner community, or to enhance the credibility and cultural relevance of the study [55, 81, 85, 89, 90, 109–112, 131]. One study in particular reported doing so as part of an Indigenous methodology, which "insist[s] that research topics and questions must reflect community needs, respond to Indigenous goals and protocols, and be collaborative" [52 (p. 463)].

With regard to data collection, some studies reported involving communities in the design of the data collection process, advising on how data ought to be collected [73, 85, 101, 113–115]. This involved, for example, communities directing whether or not to conduct individual interviews [114] and ensuring that methods were culturally appropriate [101]. In other instances, data collection tools were reviewed by communities [60, 63, 66, 92, 96, 103, 109, 115–118] to ensure, for example, that "survey questions were appropriate, coherent, and relevant" [109 (p. 69)]. In many studies, community members were also involved in recruiting study participants [e.g 61, 63, 73, 81, 92, 104, 109, 110, 115, 119, 120], and in collecting the data in the field by administering surveys, conducting focus groups, and implementing other such methods [e.g. 60, 61, 75, 83, 85, 73, 113, 118, 121–123]. Community members predominantly collected data in tandem with academic researchers, however in one case, the community led data collection on their own: "Researchers and local professionals provided training for community members on data collection methods. After this training, the communities of Nain and Rigolet directed their own local data collection" [113 (p. 95)]. While there was a relatively high number of studies that reported community engagement during data collection, far fewer involved the community in data management. Those that did, mentioned community involvement in data cleaning [59], data translation [73], ensuring accuracy of transcripts [81], and establishing a data management agreement [93].

Most studies that involved communities in data analysis reported doing so in such a way that involved seeking feedback and input into the results [e.g. 59, 70, 74, 91, 103, 115, 117, 121, 124] oftentimes to ensure accuracy and reliability of findings [e.g. 67, 70, 99, 104, 109, 117]. Relatively few reported working directly with communities to collaboratively analyze the data. Those that had community partners involved in the coding process for qualitative studies, for instance, tended to involve trained community field workers or community members on their research team [e.g. 57, 91, 98, 123, 125, 126]. Beyond analysis, four studies also noted community participation in the development of recommendations resulting from the research [55, 101, 127, 128].

Finally, with regard to dissemination, beyond sharing findings during the analysis phase, the majority of studies reported using a unidirectional approach to sharing the knowledge generated from the research, which meant that results were shared from researchers *to* the community. However, a small number of studies reported undertaking dissemination activities by working *with* the community [60, 92, 108, 112, 129]. For example, a digital storytelling project by Cunsolo et al. resulted in the Rigolet Inuit Community Government establishing the "'My Word': Storytelling and Digital Media Lab, creating the first northern Canadian centre for digital media and community-engaged research and capacity development–Inuit research and facilitation by and for Inuit" [108 (p. 133)]. Another study hosted a knowledge sharing event with the community, whereby they shared a lunch of local food which they noted as important to knowledge exchange [57].

*Capacity-building and reflexivity*. A number of articles involved research assistants from the community in data collection which offered them mentorship and skill-building opportunities [91, 98, 99]. Some articles did not refer specifically to the term 'capacity-building', however local research assistants were nonetheless provided training in relevant research skills [e.g. 80, 113, 122, 130]. Few articles mentioned training in data analysis [57, 91]. In only one instance did the local community partner (by way of a community advisory committee) develop and lead such training [80]. The majority of articles that made note of capacity-building did so in a way that was unidirectional, from researchers to community members (e.g. by training community members in research-related skills), as opposed to recognizing that communities can in turn build the capacity of researchers. Only two articles reported this kind of bidirectional capacity-building [60, 69].

With regard to reflexivity (i.e. researcher's recognition of their positionality in the co-production of knowledge), while largely present in research recommendations, the practice of it was seldom reported in empirical articles. At times, a process of reflexivity was undertaken by non-Indigenous researchers attempting to navigate research 'in a good way' [58, 60], whether in recognition of the need for decolonizing research [44], or with recognition that learning to do so is a lifelong process [58]. For specific quotes to illustrate our findings on how articles reported engaging communities, refer to Table 3.

*Recommendations*: *How should research be done and why*? *Participatory research*. Both empirical and non-empirical articles included recommendations for how research should be undertaken with Indigenous communities. A resounding sentiment in the literature was the need for the research to be participatory, and to meaningfully engage community members throughout the research process [e.g 26, 57, 99, 106, 113, 133–136]. Participation and decision-making power of the community was touted as critical for supporting the self-determination of Indigenous Peoples [11, 26, 133], and creating space for Indigenous Peoples to "ask the right questions and find their own solutions to their communities' challenges" [26 (p. 120)]. Authors of one article asserted that not only should Indigenous communities be engaged in the research as stakeholders, but they should also be recognized "as nations, not stakeholder groups- with jurisdiction over research in their communities and on their traditional lands" [26 (p. 4)]. By communities having ownership over the research, the knowledge that informs it, and its outputs, the research was described as being more respectful of Indigenous ways of being [134], more useful to the communities [85, 134], and more likely to be taken up by communities via community-led solutions [130, 135, 137]. In addition to the inherent rights of Indigenous communities to make decisions about research that is important to them, there was also a recognition that their involvement is critical to the success of research efforts, for instance by ensuring that it is culturally appropriate [57, 113, 135], and by promoting community participation [99, 136].

*Relationship-building*. One of the most resounding components of participatory research from recommendations in the literature was relationship-building [57, 70, 95, 99, 103, 130, 133, 137]. Relationship-building was discussed as the foundation to all research endeavours, and rather than being merely a 'first step' in a research project, it was deemed essential as an ongoing process [103] that extends far before and after the research takes place [137]. This was in part suggested as a way to foster trust amongst all research partners, particularly given that there have been many instances where research misconduct has bred a sense of mistrust [118]. Furthermore, building trusting and respectful relationships requires "humility, commitment, and connection' in order to engage community partners 'intellectually, socially, and emotionally" [103 (p. 282)]; flexibility on the part of researchers [99, 133]; and mutual learning in order to create ethical space for different knowledges and ways of knowing [57].

**Table 3. Illustrative quotes of community engagement from empirical articles.**

| Reference # | Quote |
|---|---|
| **Relationship Building** | |
| [85] | "The service provider staff-community relationships that existed prior to the research project were central to the establishment of trusting relationships between youth, their families, and academic team members at the start of the project. . .Specifically, it was the existing trust community members have of community partners that shaped the ways in which the project was accepted by the community and the new relationships that developed as a result" (p. 7) |
| [57] | ". . .considerable time was invested in building relationships between the researchers and team members in the community to develop trust through a shared understanding of intentions, motivations, and interests." (p. 122) |
| [52] | "Honoring and being accountable to all relationships are key tenets of Indigenous methodologies (Kovach, 2009; Louis, 2007; Wilson, 2008). Community members and researchers invest significant time and energy in building relationships and developing the trust that allows stories to be told (Kovach, 2009). We appreciate it is important to communicate some measure of the scope of the work to academic readers; however, we do so with some unease, for these metrics do not necessarily measure accountability or reciprocity on the part of the researcher, or indicate success from a community's perspective." (p. 464) |
| **Determination Over the Research Process** | |
| [95] | "As part of that effort to give authority and legitimacy to community members as knowledge producers, we created an explicit division of labor into two research sub-teams: a NunatuKavut-based team and an academic team. Over a 4-year period, from 2011 to 2015, the teams conducted their work separately and sequentially so that the community work was prioritized and the academic work was driven by the community team. Together, we refined the understanding of what was needed in a rigorous and effective system of community oversight and control of community-university research." (p. 1864) |
| [97] | ". . .due to an innovative funding model which saw all finances for this project transferred directly to the Rigolet Inuit Community Government, all research decisions were co-decided between the community government and the research team" (p. 17) |
| **Community Engagement in Specific Aspects of the Research Process** | |
| [80] | Design of research: "The CAC [Community Advisory Committee] members taught us about what is culturally relevant and appropriate in the design and implementation of the research. They reviewed the research plans and suggested modifications that reflected the unique culture and circumstances of the community. (p. 23) |
| [109] | Design of research: "Aspects of this project were community-based in nature, in that it was a priority that project goals, objectives and methods were appropriate to and developed with community members and advisors to ensure that they were inclusive in meeting community needs, while incorporating local perspectives and knowledge. . ." (p. 42) |
| [103] | Design of research: "Through the engagement of participatory principles, the community partners and I designed this study to reflect a deep respect for the intellectual and intuitive capacities of Aboriginal women. Full and active participation by Aboriginal women in the development of the research project, recruitment of participants, and modification of the discussion guide, as well as collection and analysis of data, ensured a participatory approach of mutual benefit" (p. 278) |
| [121] | Data collection: "All data were gathered by Local Research Coordinators (LRCs) in each of the five communities. Between November 2012 and March 2013, the LRCs conducted in-depth, semi-structured interviews with 17 youth (ages 15–25)." (p. 135) |
| [55] | Data analysis: "As we were wrapping up the focus group, one participant asked if the group could help with recommendations that I would be writing in chapter five. . .This I felt allowed the group to see what was written on the flip charts collectively to discuss as a group what recommendations could, or should be made." (p. 54). |
| [131] | Dissemination: "Research results were continually shared, discussed, and validated with community members to ensure accuracy and authenticity of results via regular community story nights, presentations at large community events, DVDs delivered to each household, a Facebook group, posters, and household flyers. To share research results with the academic community, researchers and community members are working collaboratively on articles and have a number of scholarly manuscripts at various stages in the peer-review process. As well, to-date, research results have been shared through 34 presentations at academic conferences by community leaders, university students, and health professionals." (pg. 97) |
| **Capacity Building and Reflexivity** | |

*(Continued)*

**Table 3.** (Continued)

| Reference # | Quote |
|---|---|
| [60] | Capacity-Building: "Our participatory and collaborative approach followed Pearce et al., (2009) and included early and ongoing communication with the NG and NGSAR, involvement of the NG in research design and development (e.g. regarding decisions on project scope, methodologies, fieldwork timing and duration), opportunities for local employment and bidirectional capacity-building (e.g. research training, co-presentation at conferences), and dissemination of results to partners and the wider community throughout the study and in a variety of mediums" (p. 49) |
| [99] | Capacity-Building: "In line with the foundations of CBPR, involving the NWG as equitable partners in the research process, from problem definition to decision making, and fostering bidirectional learning capacity to create new knowledge, was beneficial to the process." (p. 201) |
| [60] | Reflexivity: "Our approach was informed by the ongoing critical dialogue taking place in Canada and elsewhere on the need for ethical and decolonizing research and reciprocal research relationships with Indigenous communities (e.g., Castleden et al., 2012). As non-Inuit researchers, we used community engagement as a strategy to help ensure that the project addressed community needs and goals, and prioritized reflexivity." (p. 19) |
| [58] | Reflexivity: "It is no easy task as a white person to write about Aboriginal matters. It has taken about 25 years after my first contact with a Native Elder in Arizona for me to have some comprehension of a First Nations ways of being. The experience of living at Conne River not only opened the doors to understanding Aboriginal community life, it gave me a new understanding of my own roots in Judaism. I believe these roots have allowed me an innate understanding and comprehension of the Aboriginal struggle as well as a willingness to explore the subject. My understanding of traditional Native spiritual teachings derived from field experience and life." (pg. 35) |
| [132] | Reflexivity: "The use of participatory methods throughout the course of this research process was limited by my geographic location in Montreal, and the mobility of youth once they complete their high school education. At times I struggled with this sense of dislocation, but felt that it eventually contributed positively to the writing process, as I reflected on the problems and possibilities of engaging in this kind of research." (p. 208) |

*The reference numbers are listed as per the reference list

*Capacity-building.* Capacity-building was also one of the most frequently cited components of Indigenous participatory research within the literature. One suggested approach to capacity building was to provide research training to community members, and to fund community members as research staff [26, 106]. This was in recognition that oftentimes communities are balancing research as just one task amidst many other pressing responsibilities; in order for communities to be able to better control and contribute to research initiatives in their communities; and to ensure the research conducted is ethical [26]. Importantly, capacity-building was also expressed with regard to the upstream processes that facilitate research. For example, one article called for granting agencies to provide financial support directly to communities in order to recognize their ability to address their own priorities [94]. Others drew attention to the need for funding bodies to encourage researchers to offset the community-level costs of community engagement (for instance by including a budget line item for administrative overhead within communities) [95], and by recognizing, supporting, and accommodating participatory research by, with, and for Indigenous Peoples [133]. This was encouraged for researchers as well at the grant application stage, with a suggestion that they build in adequate time and financial resources to support relational and participatory approaches to ethical research practice [26].

*Ethical research.* Several articles reported recommendations for approaching ethical research with Indigenous communities, with a number identifying the need for changes at the level of research ethics boards (REBs) [11, 69, 70, 95]. For instance, Brunger and Wall [95] suggested that institutional REBs ought to assume the 'burden' of ethics review, so as to alleviate communities from the task, and allow them time to focus instead on ensuring the research

abides by OCAP™ principles and to consider whether the research is appropriate for the community. This was proposed in order to attend to principles of justice by not encumbering communities with research processes [95]. Brunger and Bull [69] also advocated on behalf of community research review processes that are distinct from REBs. The reasoning provided was to strengthen lines of accountability, to ensure community goals are respected by all involved parties and so that such a committee "explicitly attends to research in the context of ongoing colonialism" [69 (p. 139)]. With regard to suggested process changes, one article suggested that REBs revise their policies to "require the community review of research and written approval prior to REB submission and approval" [70 (p. 20)], which was in contrast to Brunger and Bull [69] who stressed that doing so then places the onerous task of identifying potential research risks on community research review committees. While relating to peer review requirements as opposed to REB, another article provided a series of indicators from which to assess ethical Indigenous research prior to publication, including the co-authorship of community representatives, community permission to conduct the research, a description of the community representative's relationship to the research, acknowledgement of community advisory board contributions, and the role of the community written into the body of the paper [45].

*Reflexivity*. Finally, article recommendations frequently alluded to or stated explicitly, the need for reflexivity on the part of researchers [90, 103, 129, 138], and emphasized the importance of valuing Indigenous worldviews [103, 132, 134, 139]. As stated by Martin et al., "Maintaining openness and reflexivity in the search for decolonizing methodologies, and actively working to educate and dismantle colonial systems" [129 (p. 116)] is important for supporting Indigenous communities in the process of "regaining their health" [129 (p. 227)]. This sentiment was also shared by Ninomiya and Pollock [90] with regard to advancing participatory research as a decolonizing methodology. Specifically, Ninomiya and Pollock [90] stated that by having researchers "write frankly about their positionality, challenges, and solutions to the lived realities of putting Indigenous CBR concepts and principles into place, the more we will further the work of decolonizing research" [90 (p. 35)]. Additionally, Loppie [103 (p. 282)] further stressed the importance of researchers accepting that they cannot always "get it right", and that the notion of "right" is not a component of all Indigenous paradigms. These tensions in knowledge systems, while accepted as inevitable by Ninomiya and Pollock [90], also proved problematic. For example, Getty [134] wrote of applying critical social theory to research with Indigenous Peoples, and explicated that when this theory is applied through a Western lens (i.e. one that privileges issues of economics and politics as opposed to spirituality, self-determination and sovereignty, among others), it could serve to further colonize Indigenous Peoples. In other words, "the choice of a theoretical framework for research with Indigenous Peoples can either contribute to their continuing colonization and oppression or their emancipation" [134 (p. 12)]. For specific quotes to illustrate recommendations in the literature for undertaking Indigenous health research, refer to Table 4.

## Discussion

This review has provided insight into the synergies and divergences of what the literature recommends as best practices for undertaking Indigenous health research, and how research is being reported, in Atlantic Canada. Several patterns emerged from our analysis. First, while the recommendations pointed to the importance of participatory, community-driven, and/or community-led research for Indigenous communities to assert their sovereignty over the research enterprise, very few empirical articles included a description of engaging the community in all aspects of the research process, even amongst those that employed participatory

**Table 4. Illustrative quotes from research recommendations.**

| Reference # | Quote |
|---|---|
| Participatory Research | |
| [11] | "...an integral shift that is required is "to meaningfully acknowledge Indigenous partners as nations, not stakeholder groups–with jurisdiction over research in their communities and on their traditional lands" [26, p. 4]. (p. 3) |
| [26] | "Most importantly, collaborative research like community-based participatory research supports both the autonomy of Indigenous Peoples and their abilities to ask the right questions and find their own solutions to their communities' challenges. "(p. 15) |
| [134] | "Community-based participatory action research provides an approach that enables Aboriginal people to participate as researchers in partnership with academic researchers. This will facilitate research that is useful to the community in which the research is conducted and respectful of their ways of being." (p. 5) |
| Relationship-Building | |
| [99] | "Building a respectful and trusting CBPR relationship goes beyond having access to a community and collecting data...it is instrumental in the process of decolonizing research in the academy. Researchers need to incorporate flexibility into their approach and timelines, and share decision making with the community" (p. 203) |
| [57] | "...it is essential to spend time developing relationships and trust among all research partners. This instils confidence in the researchers and in the project. It also addresses suspicions–fostered by previous, inappropriately conducted, research activities" (p. 124) |
| [137] | "Building partnerships with communities is a continuous process that cannot start and stop with funding." (p. 7) |
| Capacity-Building | |
| [26] | "A re-organized focus on supporting the capacity of Mi'kmaw and other Indigenous communities in research is needed so that communities can participate in and control research in their communities. In order to build that capacity at the regional and national levels, additional funding is needed..." (p. 11) |
| [94] | "Funding agencies are encouraged to enable and promote research that genuinely engages communities by providing financial support directly to communities, thereby showing respect and recognition of their abilities in addressing their local priorities and research gaps" (p. 98) |
| [95] | "...there is a need for funding to communities to offset the costs of community engagement. This could, for example, take the form of a kind of fee-for-service arrangement, similar to the arrangements that multi-site clinical trial sponsors have with university REBs. Alternatively, public funders such as the Canadian Institutes for Health Research (CIHR), the Social Sciences and Humanities Research Council of Canada (SSHRC) and the Natural Sciences and Engineering Research Council of Canada (NSERC) could encourage and fund researchers to include a line item in their budget for a 10% administrative overhead cost that would go" (p. 1873) |
| Ethics | |
| [70] | "Research ethics review board members need to revise their procedures to require community review of research and written approval prior to REB submission and approval" (p. 20) |
| [95] | "...university REBs should liaise and negotiate with communities to ensure that the burden of the ethics review falls to the REB, freeing the community RAC [Research Advisory Committee] to attend to the principles of OCAP, appropriateness of the proposed research to the community, and consideration of the existing research burden on the community in relation to identified research priorities...Although this adds a step for the researcher and the HREB [Health REB], the priority, in keeping with the ethics principles of justice, lies with lightening the burden for the community RAC." (p. 1873–1874) |
| [69] | We advocate a community review process, distinct from the "ethics" review of research ethics boards, that explicitly attends to research in the context of ongoing colonialism. Such a system places the research process in the hands of researchers in consultation with communities, with transparent and obvious lines of accountability, with appropriate oversight by research ethics review boards and, where the cohesiveness and homogeneity of the community permits, with community review committees. A good working relationship between communities and the REBs will ensure that the two distinct levels of review enable, rather than silence, good research" (p. 139) |
| Reflexivity | |

(*Continued*)

**Table 4.** (Continued)

| Reference # | Quote |
|---|---|
| [90] | "As more researchers and community stakeholders write frankly about their positionality, challenges, and solutions to the lived realities of putting Indigenous CBR concepts and principles into place, the more we will further the work of decolonizing research." (p. 35) |
| [99] | "Maintaining openness and reflexivity in the search for decolonizing methodologies, and actively working to educate and dismantle colonial systems, is the crucial role that akaneshau [white person] can play in supporting the Innu process of regaining their health." (p. 226–227) |
| [103] | "Incorporating those methods into the context of Indigenous principles requires humility, commitment, and connection. . ..Accepting our failure to always "get it right" or that the concept of "right" does not necessarily exist within many paradigms is often difficult for Western researchers" (p. 282) |

*The reference numbers are listed as per the reference list

research. Of those articles that reported some level of community engagement, the majority emphasized relationship building at the onset of the research, and community involvement in data collection. With regard to analysis, while research teams frequently brought the results back to the community for feedback, very few described involving community members in the analysis process itself. Second, and relatedly, the results revealed an evident need for greater capacity building, both in terms of enhancing the technical skills of community members, and bolstering the knowledge and skills of academic researchers to undertake projects in a culturally meaningful way. Finally, in order for research to truly respect the sovereignty of Indigenous communities, and foster self-determination, recommendations in the literature point to the need for transformations at the institutional level, so funding bodies, ethics boards, and peer review processes can better respect and promote participatory research with Indigenous communities. In practice, however, there was little mention or evidence in the published research of such transformations.

Participatory and community-driven or community-led research hold promise for altering the research praxis by reversing power imbalances and enhancing research accountability for underserved groups [140]; however, inconsistencies remain in its application [141]. This was evident in the literature reviewed and has been highlighted by other authors. In an analysis of 232 CBR research proposals, for instance, Stoecker [142] found that few projects described community members in designing the research question and analysing the data, and most involved community members during data collection, a finding that is congruent with the current study. It similarly aligns with Hackett's [143] scoping review on Indigenous youth decision-making in Community-Based Participatory Research, which found that while Indigenous youth contributed to data analysis, it was uncommon for them to participate in meaning making from the data. The reasoning for these inconsistencies may be the logistical challenges associated with community engagement in certain phases of the research over others, resource constraints, or skill gaps, and in some cases, the amount of community involvement may be determined by the capacity and interest of community members. A key tenet of participatory research is respecting community authority over research decision-making [144], and as such, due attention must be paid to the fact that appropriate, respectful, and meaningful community engagement might differ according to the unique preferences of diverse communities [34]. This is not, however, to abdicate researchers of their responsibility to ensure that decisions of if, when, and how to participate are those of communities themselves, and not based on researcher assumptions.

From our perspective, the ideal approach to participatory Indigenous health research is to have Indigenous communities governing the oversight of research that is occurring on their

territories and/or impacting their peoples. If this involves researchers who are external to those communities, then clear parameters should be defined that identify expectations around data sovereignty, community involvement, and dissemination. It is ultimately at the research design phase that these processes should be outlined and approved to determine what degree of engagement is acceptable and meaningful and to set the standards of maximum reciprocity established by the community [145].

In addition to inconsistencies of *when* community members are involved in community-led and participatory research, discrepancies exist also in *how* they are involved. Damon et al. [141], for instance, explored how peer researchers involved in CBPR studies with people who use(d) drugs perceived their experiences, and while most supported the principles of CBPR, praising it for facilitating research that is meaningful, de-stigmatizing, and non-exploitative, some participants noted their involvement as being tokenistic, and not beneficial to the community at large. Awareness of the discrepancy between researcher reports and community perceptions was elucidated by Oberndorfer et al. [52]. Although they report relationship-building with their partner community, they explain that having stated so in a manuscript does not guarantee that this relationship was reciprocal, accountable, and meaningful from the perspective of community members. To combat such a tension, community authors should, pending their permission, be included as co-authors on publications resulting from the research, so that researcher claims about their use of participatory research are legitimized, and they are held accountable to their actions.

A notable gap in the literature was the focus on capacity building, both for community members and for researchers. While authors note that community members received training in survey administration and interview skills for data collection, there was little focus on training community researchers in data analysis. Yet a critical component to Indigenous self-determination is data sovereignty, which not only includes directing research that meets their needs and collecting data, but also data management and analytics [146–148]. Walter and Suina [147] stress that in having Indigenous Peoples conduct data analyses, the pendulum towards a deficit-based narrative of Indigenous Peoples can swing to one that is embedded in Indigenous worldviews and ways of being, and thus a strengths-based narrative. In other words, by having Indigenous Peoples analyze data, research results can represent the Indigenous 'lifeworld' (the embodied realities of Indigenous Peoples shaped by social, cultural, and historical contexts) rather than being produced through (and thus privileging) the ontological assumptions of non-Indigenous researchers [147].

Capacity-building, however, can go both ways. The literature included in our scoping review gave very little mention to the concept of bi-directional capacity building, or co-learning, whereby the transfer of knowledge occurs dynamically from researchers to community members, and community members to researchers. Hacker et al. [149] have noted this bidirectional transfer of knowledge as essential for the sustainability of research outcomes and partnerships. It also reinforces reciprocity, a key tenet of participatory research [150], and mutual respect for Indigenous knowledges and ways of knowing. Indigenous Peoples have been researchers since time immemorial, collecting and analyzing information from the land, and from one another, to support and provide for the well-being of all their relations; thus, this knowledge has always existed. Furthermore, some Indigenous Peoples argue that capacity-building is needed at least as much by researchers, whereby there should be an onus of responsibility placed upon non-Indigenous researchers to learn more about Indigenous perspectives before ever entering into a research relationship with Indigenous partners. Additionally, should capacity-building be mentioned in the context of Indigenous communities, it should capture the need for greater self-determination and sovereignty over research rather than leading with the assumption that Indigenous peoples simply require more Euro-western research skills.

To ensure that communities are meaningfully involved throughout all aspects of the research process and that research processes promote bi-directional capacity-building, several recommendations suggested institutional-level changes within research ethics boards [70], funding bodies [94], and peer review processes [45]. With regard to the peer review process, Jones et al. [45] stipulated that submitted papers meet certain requirements to ensure that the research is sufficiently participatory, for instance by including a community author, and thoroughly documenting the role of communities in the research process. Street, Baum, and Anderson [151] further assert that the review process ought to be shifted altogether by broadening the definition of 'peer reviewers' to include community stakeholders, whose interest is not simply to have the paper published, but more importantly, to have the paper benefit communities. By reforming the peer review process to be more inclusive and collaborative, they posit that it may reduce alienation experienced by Indigenous Peoples forced to navigate colonial structures, such as the academy.

Further recommendations were directed towards granting agencies, for example by requesting researchers offset costs incurred to the community for community engagement initiatives [95], and by circumventing academic institutions and distributing funds directly to communities [94]. Funding initiatives such as this were not reported in our included literature, save for one article that transferred all research funds from the researchers to the community [92]. There have been increased efforts by granting agencies within Canada and elsewhere to incentivise decolonizing research, for instance through the 2012–2022 strategic plan published by the federal granting agencies in Canada [152], and the Indigenous Health Research Fund Initiative for Indigenous-led research by the Australian Government [153]. Furthermore, a conglomeration of community-based organizations in Nunavut (a territory in Inuit Nunangat, Northern Canada) has recently been granted $3.5 million to form the Nunavut Network Environments for Indigenous Health Research [154], and it is these community organizations that will hold and manage the funds, not an academic institution. While the movement toward these sorts of funding initiatives are in their infancy, they represent a critical shift in supporting Indigenous self-determination, and in 'walking the talk' of respecting Indigenous sovereignty over research.

Based on the findings of this review and the above discussion points, we outline a number of recommendations for researchers, as well as institutional bodies that facilitate research, to ensure that research is participatory, and adequately respects the sovereignty of Indigenous communities:

1. Communities should decide if, how, and when they are actively and meaningfully involved in each and all stages of the research, from a project's initial conceptualization (including selecting who or what institutions to partner with) to its completion, including publication, unless otherwise stated by the community themselves.

2. Reporting community engagement should be a requirement for peer-reviewed publications. Journals should support this reporting by requiring that authors include detailed explanations of community engagement in their methodologies, and by expanding their word limits to allow for this, if necessary. There is also merit in creating standards for reporting participatory research, similar to those available for qualitative research [155], systematic reviews [156], randomized trials [157], and other study designs.

3. To enhance the accountability of research claims in published articles, and to validate that communities were *meaningfully* involved in the research, a community representative(s), should be actively engaged and included as a co-author(s) on the paper, with consent and permission.

4. A concerted effort ought to be made on the part of researchers to ensure that community members receive training in research methods, if and when appropriate and desired, including data analysis, so that they are equipped with the tools necessary to not only contribute meaningfully to the current project, but indeed to lead their own research projects. Researchers should also be responsible for learning Indigenous ways of interpretation and the importance of language, community context, etc.

5. Conversely, non-Indigenous researchers, and academic institutions at large, must recognize the inherent knowledge that Indigenous Peoples possess, and how that knowledge (mostly unacknowledged) has not only informed Western science, but when respected and valued, can also contribute deeply and meaningfully to the research process, and to the betterment of researchers themselves.

6. Peer review processes, ethics boards, granting agencies, and journals should implement concrete mechanisms that (a) ensure inclusiveness of Indigenous communities in Indigenous health research, (b) privilege Indigenous knowledges and ways of knowing, and (c) support and facilitate research that is both owned and led by Indigenous communities.

7. A checklist of community engagement should be developed where researchers and Indigenous communities can collaboratively assess how the community would like to participate in the research prior to it taking place. It can then be used as an evaluative tool to hold researchers accountable while responding to the unique preferences of communities.

There are several limitations to this scoping review. First, for reasons of feasibility, we were unable to connect with individual authors to fill in any reporting gaps that existed in the articles reviewed. Thus, there may be instances when researchers omitted details about their level of community engagement. This may be because these approaches were indeed not pursued, because the researchers left this information out of the publication (e.g. if their focus was on the results rather than the methodology), and/or because of restrictive journal requirements that disallowed the inclusion of such nuance. It is also important to acknowledge that our intent was not to assess definitively whether a study was or was not participatory, but rather, assess whether and how the study reported this within the peer-reviewed literature. This is an important distinction, since it acknowledges that responsibility for reporting levels of community engagements rests not solely with the authors of studies, but also with the editorial decisions made by journals as to whether or not Indigenous engagement is a necessary reporting requirement. Second, though we assessed levels of community engagement within research undertaken between 2001–2020, we did not explore differences in how community engagement was conceptualized over that time period. Understandings of what meaningful community engagement is may have evolved over time, however we did not assess this evolution in our review.

Third, it is important to acknowledge the limitations of Western-based literature reviews in fully capturing Indigenous knowledge. According to Mi'kmaq scholar, Dr. Marie Battiste, conducting a literature review on Indigenous knowledge is "an oxymoron because Indigenous knowledge is typically embedded in the cumulative experiences and teachings of Indigenous peoples rather than the library" [158] (p. 2)]. We therefore recognize that in some ways, we are critiquing the very Western scientific approach we are assuming. However, we hold that, despite its imperfections, a scoping review provides a necessary snapshot of a broad body of work, so we can identify areas for further inquiry and improvement. Finally, while our focus was on the peer reviewed literature in academic databases, there are likely examples of participatory projects that were not submitted to peer reviewed journals, and thus not included in this review. Community-based organizations, for instance, may very well be undertaking such

initiatives, but choose not to document their approaches in manuscripts for publication and academic consumption. Despite these limitations, however, this scoping review offers a unique lens to analyze how Indigenous health research in the peer reviewed literature has reported its engagement with communities, and how that engagement aligns with recommendations for undertaking Indigenous health research. If engagement with Indigenous communities is deemed an essential aspect of undertaking Indigenous health research, but it is not a requirement for publication within peer-reviewed literature, then there is an academic imperative to improve the reporting of Indigenous health research writ large.

## Conclusion

Focusing specifically on the Atlantic region of Canada, this scoping review offers insight into whether and how Indigenous health research employs community-based participatory, community-driven, and/or community-led research methods by assessing how research teams engaged communities throughout the research process. Our findings note that although participatory methods are cited as the preferred methods to engage with Indigenous communities, there are discrepancies in the degree to which communities are engaged in the research that impacts them. This suggests that there is further work to be done to ensure that what constitutes participatory research–including community-driven and community-led processes–are upheld in instances where it is being claimed as a methodology, and that there is room to expand the level of community engagement in research across all methodologies. Although institutional and funding restrictions may play a role in the degree to which research engagement happens, there also appears to be room for research teams to deepen their commitment to participatory research. Indeed, as Indigenous communities continue to rebuild and reclaim their sovereignty over their identities, territories, and data, the expectations surrounding participatory research ought to be construed as a *baseline* measure of researcher commitment rather than a panacea. Ultimately, researchers should aim to far exceed this baseline by ensuring that the research they undertake not only engages Indigenous communities, but that it is owned and led by them. Reconciliation requires Western-based research to unlearn colonial practices of research, re-learn, and accept the inherent value of Indigenous-led research, and we assert that community-directed research is a good place to start.

## Supporting information

**S1 Table. Search terms used in the electronic databases to identify articles for the scoping review of Indigenous health research conducted in Atlantic Canada from 2001-June 2020.** (DOCX)

**S2 Table. Data charting form template.** (DOCX)

**S1 File. Reference list for all records included in analysis.** (DOCX)

## Author Contributions

**Conceptualization:** Ashlee Cunsolo, Margot Latimer, Jane McMillan, John R. Sylliboy, Debbie Martin.

**Formal analysis:** Kathleen Murphy, Karina Branje, Tara White.

**Funding acquisition:** Ashlee Cunsolo, Margot Latimer, Jane McMillan, John R. Sylliboy, Debbie Martin.

**Investigation:** Kathleen Murphy, Karina Branje, Tara White, Debbie Martin.

**Methodology:** Shelley McKibbon.

**Project administration:** Kathleen Murphy, Karina Branje, Tara White.

**Supervision:** Ashlee Cunsolo, Margot Latimer, Jane McMillan, John R. Sylliboy, Debbie Martin.

**Visualization:** Kathleen Murphy, Karina Branje, Tara White.

**Writing – original draft:** Kathleen Murphy, Karina Branje, Tara White, Debbie Martin.

**Writing – review & editing:** Kathleen Murphy, Karina Branje, Tara White, Ashlee Cunsolo, Margot Latimer, Jane McMillan, John R. Sylliboy, Debbie Martin.

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
