## [Decision Letter · Decision Letter 0]

27 Apr 2021

PONE-D-20-38099

Are we walking the talk of participatory Indigenous health research? A scoping review of the literature in Atlantic Canada

PLOS ONE

Dear Dr. Martin,

Thank you for submitting your manuscript to PLOS ONE. After careful consideration, we feel that it has merit but does not fully meet PLOS ONE’s publication criteria as it currently stands. Therefore, we invite you to submit a revised version of the manuscript that addresses the points raised during the review process.

We look forward to receiving your revised manuscript.

Kind regards,

Andrew Soundy

Academic Editor

PLOS ONE

Additional Editor Comments:

I agree with the comments made by experts.

These are comments from an expert in the area in case not below:

GENERAL COMMENTS

- More straightforward writing could be used throughout.

- Although this scoping review was focused on Atlantic Canada, it does not standout in this way. It could be from anywhere. Why is this important for Atlantic Canada? It is possible by including an aggregate descriptive table of the studies conducted in Atlantic Canada, that something distinct might be highlighted to include in the discussion and results. Nevertheless, consideration of the recommendations in this scoping review by researchers and community should expand beyond the region.

ABSTRACT

- Line 30-32, Introduction: If you present findings for ‘b’ first then switch (a) and (b).

INTRODUCTION

- More straightforward writing needed throughout. Introduction should be more succinct. A lot of content that is better suited for the discussion. Very interesting ideas, but a lot of ideas with insufficient elaboration on some and too much on others. Described below suggested shifting some text to discussions or writing more succinctly.

- Be mindful of generalizations that supports pan-Indigenous notions. ‘Indigenous worldviews are grounded in the concepts of …’ versus writing ‘Some Indigenous worldviews…’ given that there is only one reference.

- Inuit means people so ensure that sentences do not read ‘Inuit People(s)’ which might read as ‘People Peoples.’

- Although reference 5 is an interesting article about the involvement of Elders in HIV CBR more suitable references for this statement are needed.

- Line 64: ‘…complex inquiry of methods;’ include references.

- Line 79: references needed.

- Is participatory research being discussed as part of CBR, CBPR or even PAR? If so then participatory research may involves some or all of its principles and may not involve a shift in power. Describe or define your understanding of participatory research and support this with the literature.

- Line 86: ‘…community-driven methods…’ was the word ‘methods’ supposed to be the word ‘research’. This is the first mention of methods.

- Line 89: ‘…in situated ways...’ what is meant by situated ways and does the sentence hold the same meaning if this is removed.

- Line 91: ‘…participatory methods…’ are you intending to talk about a part of the research or simply the methods. Clarify.

- Line 96: ‘…Western-based research paradigm.’ References needed.

- Line 106: ‘…, and/or organization.’ References needed. What does the term participatory imply that findings and [others] would be vetted by an Indigenous group? Earlier you described that there is a range of ways community could be involved. If this is your perspective leave it to the discussion where your perspective might be supported by your findings.

- Line 113: What are the principles of participatory engagement?

- Line 115: ‘…their communities…’ references needed.

- Line 115-126: Insufficiently referenced. This sounds like you are drawing conclusions or making inferences from your experiences or observations. Most of the information in this paragraph would be better suited for a discussion with some references to support it. Find an alternative and more succinct way to connect to the next paragraph which also includes several sentences that need to be referenced (line 127-128, line 128-132) and made more succinct or would be better suited for a discussion.

- Line 128: ‘Therefore, the other point…’ this sentence seems disconnected from the first sentence. Is it about capacity mentioned in the prior paragraph? I can see how it works to connect to the ACADRE and IMNPs but should be made to be more fluid.

- Line 149: ‘…meet the needs of their communities.’ References needed.

- Line 147-168: Insufficiently referenced.

- Line 167: Does Two-Eyed Seeing enhance the decolonizing goals of community-driven, participatory research? How? If this is a perspective leave it for the discussion or cite seminal literature by Elders Albert or Murdena Marshall with/without Dr. Bartlett.

- Line 170: align objectives with the abstract or align abstract objectives with the ones in the introduction because they do not convey the same message. Use similar wording and ensure that you will clearly be examining community engagement.

METHODS

- More straightforward writing needed.

- Line 178-182: Remove the definition of scoping review, not necessary. Stick to specifying the details of the review process described in lines 183-184.

- Line 184: ‘It is important to acknowledge …’ include in the limitations section of a discussion.

- Line 203: Is the review team the authors? If so say ‘We worked with …’

- Line 204: Did the broad keyword terms fall under the umbrella of the concepts of interest? If so, connect this.

- How was the ‘Native Friendship Centres’ included in the search strategy? What does this mean?

- Line 221-223: Unnecessary to include the information in parenthesis for the database. For example, remove EBSCO, Elsevier, etc….

- Line 226-229: No need to include the search dates except for the last common search date (June 2020).

- Table 1 could be a supplementary table.

- Polish sections. For example, ‘… the following inclusion criteria if the study(ies): (1) included a focus…’ (line 241). There are multiple studies so no need to write ‘study(ies)’ just write studies. Also, this is a list of inclusion criteria so no need to write ‘included….’

- Line 245: Exclusion criteria are also not the opposite of inclusion criteria ‘…did not draw conclusions relating to Indigenous Peoples in Atlantic Canada.’ Could be removed because we already know you are going to exclude this based on your inclusion criteria.

- Line 247-249: Be more succinct.

- Line 250-252: Be more succinct. You have already written about title and abstract screening and following with full-text screening information, yet you have started this sentence by writing about title and abstract screening. Just start with full-text screening.

- Add a sub-heading for the ‘charting and analysis’ section.

- Line 290: ‘We understand that the very…’ if you remove the word very does it hold the same meaning.

- Line 302: Figure 1 is listed, but on page 23 of 68 is written figure 3. Have you inserted where to include all figures or only figure 3? Be consistent. Also, I do not see a figure 3 in your submission.

RESULTS

- Line 302: Use PRISMA for scoping reviews: http://prisma-statement.org/Extensions/ScopingReviews

- Line 314: ‘92’ initially seems like a reference. Write to the first decimal place and use ‘n’ (n=92, 54.6%) and do so for all other numbers.

- Line 319: ‘…reported engaging…’ vs ‘…reported engagement …’. Engagement sounds better.

- Table 3: ‘All of the aspects’ vs ‘All aspects’ align with heading in 4th column. Verify numbers with the table with the numbers in the text. In the table n=92 is 50.53% whereas in the text n=92 is 54.55%. ‘Combination 2 or more’ should also be more aligned with all other column headings. For example, the table title includes the word engagement and so the column headings could be: All aspects, Two or more aspects, One aspect, None. ‘CBPR stated’ no need to say ‘stated’. ‘used’ in the last row no need to include this. Clean up the table.

- Figure 2: This figure is about engagement therefore leave this in the figure title and remove from x axis labels. For example, x axis labels: All aspects, Research design, Data collection, Data analysis, Development of research question, Dissemination of findings, None. Ensure that there is no ‘…’ in the labels because it is difficult to determine what some of the ‘…’ are because they are not all written in the text. There should be no ‘…’ in figure labels. ‘Stages of the Research Process’ also sounds better than ‘Research Stage’. In general, improve the esthetics of the figure and consider changing the color of the bars from blue to black because it will likely be printed in black and white.

- Consider changing ‘in a number of ways’ to ‘in several ways’ to diversify language a bit. Some wordsmithing and proofreading: ‘prior to the initiation of the research’ vs ‘prior to initiating the research’.

- Line 353: no need to write ‘etc’ since you have already written ‘e.g.’ for example so we know as a reader that the list is incomplete and in theory more could be added. Use ‘etc.’ sparingly.

- Line 378: Spell out OCAP upon first use since the audience for this journal may not be familiar with OCAP which is also trademarked so ensure this is indicated. CBPR should also be spelled out upon first use.

- Example of being more succinct with changes included in line 378: ‘Yet, with the breadth of articles that used OCAP principles in undertaking ethical Indigenous health research,35,39,64,70,74-76,82-91 only one study explicitly gave recognition to communities as ‘owners’ of the research enterprise.’ The following sentence includes ‘… grant proposal in collaboration with one another.’ ‘with one another’ could be deleted without losing the meaning of the sentence. Also, this sentence is very long so split it in half.

- Another example of writing in a more straightforward way, line 391: ‘Studies engaged communities in several aspects of the research process ….’ The following sentence could also be written in a more succinct way.

- Some questions I had about the research was some basic descriptive information of the studies. Given there are a lot of studies just some basic aggregate information would have been interesting such as the number of studies per province (4 or 5 regions by separating NFLD and Labrador), specific Indigenous groups, quantitative or qualitative or both for example.

DISCUSSION

- Line 644: ‘The literature review…’ are you referring to your literature review. If so write ‘Our literature review…’

Journal Requirements:

2. Please include a table summarizing the characteristics of studies included.

3. We noted in your submission details that a portion of your manuscript may have been presented or published elsewhere.

"Our research team currently has a different manuscript submitted to another journal from this project. This manuscript and the current manuscript being submitted to PLOS ONE are entirely separate entities; viewing and analyzing the data from different perspectives. The first manuscript focuses on the general trends (number of articles, topic areas, etc.) of Indigenous health research within the Atlantic region and is largely quantitative in nature, whereas the current manuscript being submitted to PLOS ONE is examining the ways in which community engagement has transpired, and the recommendations surrounding the ways in which community engagement should be occurring within Indigenous health research in the Atlantic region. Notably, the first manuscript does not need to be published in order for this current manuscript to be published, as they examine different aspects of the data from entirely different perspectives. "

Please clarify whether this publication was peer-reviewed and formally published. If this work was previously peer-reviewed and published, in the cover letter please provide the reason that this work does not constitute dual publication and should be included in the current manuscript.

6. We note that you currently have two different versions of tables 1 and 2 included in your manuscript, so that the tables can be differentiated can you please update the table titles (numbering) and in-text citations so that the second set of tables are numbered individually (and not the same as the previous numbers for Tables 1 to 2 already used).

Reviewers' comments:

Reviewer's Responses to Questions

**Comments to the Author**

1. Is the manuscript technically sound, and do the data support the conclusions?

Reviewer #1: No

2. Has the statistical analysis been performed appropriately and rigorously? 

Reviewer #1: N/A

3. Have the authors made all data underlying the findings in their manuscript fully available?

Reviewer #1: Yes

4. Is the manuscript presented in an intelligible fashion and written in standard English?

Reviewer #1: Yes

5. Review Comments to the Author

Reviewer #1: 1) Abstract, Lines 28-29: The statement “However, there are inconsistencies in how it is applied in practice, as well as how it is reported.”: This statement implies that the inconsistencies of participatory research is a bad thing but I would disagree with this presumption as the application of participatory research should and would vary based on numerous factors such as prior relationships, history, nature of the research, etc. Further, this orientation to participatory research -- being flexible and in context -- is reflected in what the authors discuss on page 4, lines 87-91, which makes the message of this manuscript a bit confusing. Truthfully, how a research project is conducted with an Indigenous community, is between the community and the outside researchers. If the partnership works – albeit doesn’t check off all of the boxes for gold standard participatory research – then that is OK. Clearly, not all Indigenous communities have the interest or resources to take on all of the responsibilities outlined.

2) Abstract lines 28-29: With respect to the second point of this statement (inconsistencies on how it is reported) may warrant the creation of a reporting template as you see for other types of studies (qualitative studies: COREQ; RCTs: CONSORT).

3) Abstract, Line 38: Please note that the word “data” is plural so replace “was” with “were” in its use (abstract, line 38)

4) Abstract, Line 45: Unclear what “transformations at the institutional level” is about. Vague language.

5) Overall, if the intent of this scoping review is not what is perceived from the language used in the abstract (see comment #1), then the introduction should be revised accordingly (i.e., not to suggest that that manuscripts reporting on participatory health-related research with Indigenous communities should be uniform and consistent across the board).

6) Concerned that this study is attempting to answer questions (whether or how) solely based on the literature while how Indigenous health research is conducted does not always find itself in the literature nor does the literature about such studies offer the degree of detail with respect to the participatory elements. That is, not reporting in a research article whether/how the Indigenous community was engaged in the study does not mean the Indigenous community was not engaged (e.g., lines 316-317). Thus, the tool of a scoping review appears to be inadequate to answer the whether or how research teams are actually enacting participatory and/or Indigenous-led processes. In fact, the irony is that the authors of this scoping review are criticizing the Ivory tower / Western ways while taking an Ivory tower / Western way approach itself. Thus, the overall message of this article is very confusing.

7) While the rigor detailed in the scoping reviews methodology is strong, the framing of this study needs to be revised in light of the concerns mentioned.

6. PLOS authors have the option to publish the peer review history of their article (what does this mean?). If published, this will include your full peer review and any attached files.

Reviewer #1: No

---

## [Author Response · Author response to Decision Letter 0]

10 Jun 2021

Thank you for the very detailed and thorough review of the manuscript. We have considered each and every suggestion and have included our responses in a Response to Reviewers that outlines how we have addressed the requested edits in the revised manuscript. As we have outlined in the cover letter to the Editor, in addressing many of the bigger comments relating to succinctness of writing and questions regarding the intent of the article, the process of editing addressed many of the more minor comments. We have indicated the updated line numbers where each edit is made. Also, note that the 'manuscript with track changes' is set to 'simple markup'. If you wish to see the extensive edits made, please switch the review mode to 'all markup'.

---

## [Decision Letter · Decision Letter 1]

14 Jul 2021

Are we walking the talk of participatory Indigenous health research? A scoping review of the literature in Atlantic Canada

PONE-D-20-38099R1

Dear Dr. Martin,

We’re pleased to inform you that your manuscript has been judged scientifically suitable for publication and will be formally accepted for publication once it meets all outstanding technical requirements.

Kind regards,

Andrew Soundy

Academic Editor

PLOS ONE

Additional Editor Comments (optional):

Reviewers' comments:

Reviewer's Responses to Questions

**Comments to the Author**

1. If the authors have adequately addressed your comments raised in a previous round of review and you feel that this manuscript is now acceptable for publication, you may indicate that here to bypass the “Comments to the Author” section, enter your conflict of interest statement in the “Confidential to Editor” section, and submit your "Accept" recommendation.

Reviewer #1: All comments have been addressed

2. Is the manuscript technically sound, and do the data support the conclusions?

Reviewer #1: Yes

3. Has the statistical analysis been performed appropriately and rigorously? 

Reviewer #1: N/A

4. Have the authors made all data underlying the findings in their manuscript fully available?

Reviewer #1: Yes

5. Is the manuscript presented in an intelligible fashion and written in standard English?

Reviewer #1: Yes

6. Review Comments to the Author

Reviewer #1: The authors have provided substantial revisions and have fully addressed my comments provided at the initial review. This revised manuscript will make a good contribution the literature.

7. PLOS authors have the option to publish the peer review history of their article (what does this mean?). If published, this will include your full peer review and any attached files.

Reviewer #1: **Yes: **R. Turner Goins

---

## [Editor Report · Acceptance letter]

19 Jul 2021

PONE-D-20-38099R1 

Are we walking the talk of participatory Indigenous health research? A scoping review of the literature in Atlantic Canada 

Dear Dr. Martin:

I'm pleased to inform you that your manuscript has been deemed suitable for publication in PLOS ONE. Congratulations! Your manuscript is now with our production department. 

Kind regards, 

on behalf of

Dr. Andrew Soundy 

Academic Editor

PLOS ONE